# Glacio-archaeological evidence of warmer climate during the Little Ice Age in the Miyar basin, Lahul Himalaya, India

Rakesh Saini<sup>1</sup>Milap Chand Sharma<sup>2</sup> Sanjay Deswal<sup>3</sup>Iestyn David Barr<sup>4</sup> Parvendra Kumar<sup>1</sup> <sup>1</sup>Department of General & Applied Geography,Dr. Harisingh Gour Central University, Sagar, 470003, India <sup>2</sup>Centre for the Study of Regional Development, Jawaharlal Nehru University, New Delhi, 110067, India <sup>3</sup>Department of Geography, Government College Dujana, Dujana, 124102, India <sup>4</sup>School of Natural and Built Environment, Queen's University, Belfast, BT7 1NN, United Kingdom

*Correspondence to*: Rakesh Saini (rakesh83jnu@hotmail.com)

#### Abstract.

5

- Impressive glacio-archaeological evidence is described from the *Miyar* basin, Lahul Himalaya, India. Three ruins, namely *Tharang, Phundang* and *Patam* are identified along with evidence for past settlement and rich irrigation practices in the basin. These ruins are located in the end moraine complex of *Tharang* glacier, just ~2–3 km from the present glacier snout. Reconstruction of these ruins was undertaken based on mapping and radiocarbon (<sup>14</sup>C) dating. The radiocarbon dates (9 samples were dated) indicate that the settlement was occupied between cal AD ~1170 and cal AD ~1730, thereby encompassing the majority of Little Ice Age period. The settlement's occupation at ~3700 m a.s.l. (whereas present habitation is restricted to areas below ~3500 m a.s.l.) for almost ~550 years during the 12<sup>th</sup> to 17<sup>th</sup> centuries suggest warmer
- conditions than today. Moreover, the study finds no evidence to suggest any noticeable glacier advance during this period.

# 1 Introduction

- 20 The style and timing of past glacier fluctuations in the Himalaya is widely contested. However, the Late Quaternary maximum glacier expansion in the Himalaya is known to have been asynchronous with the global Last Glacial Maximum (Barnard et al., 2004a; Gillespie and Molnar, 1995; Goudie, 1984; Hedrick et al., 2011; Lehmkuhl and Owen, 2005a; Murari et al., 2014; Owen et al., 1996a, 1998a, 2001, 2002a ; Sharma and Owen, 1996; Solomina et al., 2015; Taylor and Mitchell, 2000; Wanner et al., 2008a). Likewise, it is now widely acknowledged that the timing and extent of Holocene climatic
- variability, and associated glacial fluctuations, in the Himalaya differs notably from more northerly latitudes (Clapperton and Sugden, 1988; Dean et al., 1984; Gliganic et al., 2014; Lehmkuhl and Owen, 2005a; Mayewski et al., 2004a; Murari et al., 2014; Owen, 2009; Owen et al., 2002a; Schaefer et al., 2009; Solomina et al., 2015; Steig, 1999; Wanner et al., 2008a; Xu et al., 2015). However, conditions during the Little Ice Age (LIA) reported from other parts of the world (Bradley and Jones, 1993; Mann, 2002) have been generalised over the Himalaya (Barker, 2007; Goudie, 1984; Holloway, 1997; Mayewski and
- 30 Jeschke, 1979; Mayewski et al., 1980; Owen et al., 1996a, 1998a, 2000; Rowan, 2016; Taylor and Mitchell, 2000). Generally, in many parts of the world, the LIA is considered to have occurred from~1300 to 1850 AD, and resulted in notable glacier expansion and lower temperatures than today (Bradley and Jones, 1993; Mann, 2002). Despite these general descriptions, climatic conditions and associated glacial activities during this period remain topics of debate at global and

5

regional scales. For example, some studies have reported that during the LIA, glaciers throughout the world expanded (Grove, 2001; Porter, 1981), while others point out that glacier expansion only occurred in areas north of 20°N (Mann, 2002; Matthews and Briffa, 2005). In addition to such disputes, researchers sometimes have differing opinions on the causes of LIA glacial expansions. For example, some consider LIA glacial expansions the consequence of increased precipitation (Bradley and Jones, 1993), whereas, others suggested that this was driven by a significant fall in temperature (Grove, 2001; Hughes and Diaz, 1994; Mann, 2002; Nesje and Dahl, 2003). Such uncertainties often persist because of a general absence of LIA records from different parts of the world, limiting our ability to assess regional and global patterns of climate and glacier behaviour during this period.

At a regional scale, existing records suggest LIA glacial advance in adjoining and different parts of the Himalaya (Owen et al., 2000; Derbishyre and Owen, 1997; Hedrick et al., 2011; Lehmkuhl et al., 1998a; Murari et al., 2014; Owen and Derbyshire, 1998; Owen et al., 1996, 1998b, 2001, 2002a, 2002b, 2005; Sharma and Owen, 1996; Solomina et al., 2015; Spencer and Owen, 2004; Taylor and Mitchell, 2000). However, these records vary in terms of the timing and extent of glacial advance they propose, and most are based on relative rather than absolute chronologies of events. Wherever absolute chronologies have been obtained (Barnard et al., 2004a, 2004b, Owen et al., 2001, 2002b; Seong et al., 2007a; Sharma and

- 15 Owen, 1996; Tsukamoto et al., 2002), most are based on techniques including thermo luminescence (TL), opticallystimulated luminescence (OSL) and cosmogenic radionuclide (CRN) dating. Though these methods are widely applied globally, they have associated limitations (Bailiff et al., 2014; Jensen et al., 2000; Blair et al., 2005; Spencer and Owen, 2004; Wallinga, 2002). Given the limited availability of organic material in recently deglaciated environments in the Himalayas, radiocarbon (<sup>14</sup>C) dating has been used less frequently. However, where datable organic material can be found,
- <sup>14</sup>C dating potentially provides a means of establishing a robust chronology for recent glacial and climatic fluctuations in the higher Himalaya. Therefore, this study presents a chronology (based on <sup>14</sup>C dating) of archaeological remains (ruins) full of datable organic material, in the form of charcoal, wood, horn, hearth soil, bone, and cowshed soil. The study is helpful in allowing a greater understanding of the relationship between glacial fluctuations and human occupancy in this area during LIA. Furthermore, this study provides a framework to understand the spatial character of complex climatic variability in the
- 25 North West Himalaya during the LIA.

## 2 Location of the study area

The *Miyar* basin is a major sub-basin of the lower Chanderbhaga valley within the Great Himalayan Tract of the Lahul and Spiti district of Himachal Pradesh, India (fig. 1.). The basin extends from  $32^{\circ}42'36''$ N to  $33^{\circ}15'24''$ N and  $76^{\circ}40'12''$ E to  $77^{\circ}1'15''$ E, with an area of ~963.85 km<sup>2</sup>, out of which ~21.6% (208.2 km<sup>2</sup>) is covered by glaciers. The basin contains a

30 cluster of 92 glaciers of varied dimensions and types out of which *Miyar* glacier is the longest (~27 km). Due to its location between Pir-Panjal to the South and the Great Himalayan range to the North, the basin, with an altitudinal range of ~3800 m (~2600 m to 6400 m) lies in a rain shadow zone of the South West Monsoon. As a result, the area is classified as a cold desert. Climatically, the area is dominated by severe and long winters that last from September to May. During winter

5

months, temperatures fall to -35°C. However, during summer months (May-September), day temperature rises to 35°C but can fall as low as -2°C during night or on any cloudy day. The average precipitation is 679 mm yr<sup>-1</sup>, and is dominated by snowfall. The permanent snowline is located between 4900 and 5200 m a.s.l. The area above 4000 m is dominated by periglacial processes, such as freeze-thaw, shattering, solifluction etc., whereas altitudes between 3500 and 4000 m are dominated by paraglacial processes. Only the area below 3500 m is currently used for cultivation, but the higher reaches are used for grazing as alpine pastures during summer months.

## 3 Methodology

### 3.1 Mapping

Extensive mapping was carried out of the *Tharang* end-moraines complex and the ruined settlement within its confines. The
mapping included delineation of glacial, glaciofluvial, periglacial and paraglacial landforms, and shattered walls of the
former buildings on each mound. The mapping techniques are elaborated below.

# 3.1.1 Field Mapping

In 2008, plain table surveys, calibrated with clinometers, compasses and GPS, were carried out at 1:10000 scale within the

- confines of the lateral moraines. In the same year, we began to monitor the terminus of *Tharang* glacier using a hand-held Garmin-76CS GPS. In addition, a DGPS (Trimble-GeoXT) survey was carried out, installing a base station within the *Tharang* glacier forefield (at 3600 m a.s.l.) for continuous GPS monitoring, with continuous power supply using a Honda-Poratble Genset for the base. The rover was taken up to the terminus (~4471m a.s.l.) of *Tharang* glacier. The Terminus and the proglacial area, along with lateral moraines, were mapped using point, line and area feature classes. Between 2010 and
- 2012, ruins were identified, and their irrigation systems (*Kuhls*) and agricultural fields were mapped using the GPS (76CS), along with detailed photographic recording of relevant features. In 2014, mapping of the shape, size and association of ruins was completed using plain table and chain-tape survey methods, calibrated with the handheld GPS. Based on the above analysis, these ruins were identified as dwelling settlements, with common rooms, kitchen rooms, store rooms, cow sheds and places of worship. Mapping of the end moraine complex, water-harvesting bodies, agricultural fields and irrigation
- systems was finally completed in 2015, using the Robotic Total Station (Trimble), calibrated with Juno-5 3D GPS. In the same year, we collected 11samples for radiocarbon dating of organic material extracted from these remnants.

#### 3.1.2 GIS Mapping

Field data were incorporated into ESRI Arc GIS 10.1 software by integrating post-processed GPS/DGPS, plain table and total station data. Post-processed hand-held GPS data using GARMIN MapSource and DGPS rover and base data in the Trimble Pathfinder software were integrated into mapped shapefiles. The topographical sheets of Survey of India were used for mapping the basic location of features in the basin, whereas high-resolution ESRI Arc GIS online base maps were used for detailed vectorisation of the *Tharang* end moraine complex and ruins. Google Earth 3D visualization of terrain between

the scales of 0.1 and 3.0 vertical exaggeration was also used for mapping geo-archaeological features, moraines, hummocks and deglaciated valleys. In addition, each of the settlements, agricultural fields, temples, water-harvesting ponds and irrigation channels (*kuhl*) were identified and mapped from these sources, wherever possible.

# 5 3.2 Radiocarbon Dating

#### 3.2.1 Sample collection

We collected 11 samples for radiocarbon dating in the form of charcoal, wood, horn, hearth soil, bone and cowshed soil from the ruins, out of which 9 samples were dated (other samples contained insufficient organic material). Initially, based on settlement measurements and their relative locations, we identified common rooms and kitchens within the ruins. According

to architectural practices in *Lahul*, common rooms are relatively large, contain a hearth, and are able to accommodate all family members. In these common rooms, we identified 6 hearths which had preserved wood, charcoal, and bone samples for subsequent dating.

#### 3.2.2 Laboratory Work

- Radiocarbon samples were processed following Reimer et al., (2015) at the 14CHRONO Centre of Queen's University Belfast, Northern Ireland, United Kingdom. Samples were processed at three stages; namely pre-treatment, combustion and graphitization, and finally determining age using Accelerator Mass Spectrometry (AMS). The samples were cleaned using hydrochloric acid, sodium hydroxide, gelatinization and ultra-filtration depending upon the nature of the sample (bone, wood, charcoal, and sediment) in the pre-treatment stage to remove the exogenous carbon. Thereafter, these treated samples
- were dried and converted into carbon dioxide and then to graphite at combustion and graphitization followed by processing in the AMS. The obtained radiocarbon dates were calibrated in the CALIB RADIOCARBON CALIBRATION PROGRAM 1986-2016 using IntCal13 radiocarbon age calibration curves 0-50000 years cal BP in conjunction with Stuvier et al. (1993), Reimer, (2013). The calibrated ages were recorded with one sigma ranges.

#### 25 4 Results

# 4.1 Glacier Advances and Landforms

Geomorphological landforms such as lateral moraines, recessional moraines, hummocks, outwash plains, lacustrine deposits, erratic and drumlins illustrate the region's glacial and associated climatic history. Though, in the present study, we focus on the late Holocene glacial records and climatic conditions during the presumed LIA, along with human activities within the

30 moraine complex of *Tharang* (a tributary of *Miyar*), the most prominent glacial advance, reflected by large U-shaped trunk and tributary valleys, scoured shoulders and bedrock walls and truncated spurs, occurred during the *Miyar* Glacial Stage (Saini, 2012, Deswal, 2012). This stage still remains to be dated due to the paucity of datable material. Considering the size and extent, probably the stage is contemporaneous with the Chandra Stage proposed by Owen et al., (1996b, 2001) in the upper Chanderbhaga valley. During this period, *Miyar*, the largest glacier in the region, fed by the tributary glaciers,

expanded down-valley ~40 km to reach Karpat, with an ELA depression of ~400 m, relative to the present. This glacier left behind a deep box-cut upper valley of ~200 m depth, scoured valley walls and truncated spurs.

Following deglaciation from this maximum extent, the next glacial advance in the basin has been dated to ~6.6 ka, based on OSL dating of sediments from the end moraine complex of *Tharang* glacier (Deswal, 2012). During this mid-Holocene advance, *Tharang* glacier expanded to reach ~3595 m a.s.l., in comparison to its current terminus position at ~4470m a.s.l. This represents a vertical displacement of ~880 m in a distance of just ~4.95km (fig. 2.). This advance produced the highest lateral moraine in the *Tharang* complex, marked as LM-I (lateral moraine) on the left and right flank of the *Tharang* channel (fig. 3.). Within the area up-valley of LM-I, there are three sequential lateral moraines marked as LM-II, III, and IV, on the left flank of the *Tharang* channel, which sequentially reduce in relief (fig. 3.), denoting a progressive history of the retreat of

- Tharang glacier. Though smaller than LM-1, the volume of sediment deposited at lateral moraines II, III and IV suggest that, at times, retreat was slow enough (perhaps including still-stands) to form extensive moraines, or that the glacier experienced small stages of re-advance during overall retreat. However, the well-marked 23 recessional moraines on the right flank of the *Tharang* channel (fig. 3.) suggest that after the middle Holocene advance the retreat had been progressive.
  Following the middle Holocene glacier advance, the next notable position of glaciers in the basin can be identified on the
- basis of sediments and landforms within a kilometre from the present glacier termini (Plate 1). These sediment dumps are clearly distinct from the middle Holocene moraines, as they lie ~2–3 km further up-valley. Moreover, the middle Holocene deposits have significant lichen cover which is missing on deposits close to the present glacier termini. Further evidence to constrain the glacier margins from historical surveys is limited. However, the former terminus position of a few glaciers, including *Tharang*, were inadvertently incorporated into the Great Trigonometric Survey (GTS, 1870) map (fig. 4.) and later,
- the map reproduced by captain Harcourt in 1871(Harcourt et al., 2010). It seems that these deposits belong to the Historical Advance (between 18<sup>th</sup> and 19<sup>th</sup> century) which can be explained by the strengthened Indian Summer Monsoon around the 18<sup>th</sup> century (Sinha et al., 2011; Thompson et al., 2000). In comparing the length change of glaciers during this period, we find that, within the entire basin, Chhudung glacier experienced the largest advance of ~2.50 km, with its terminus extending down to ~3930 m a.s.l. (this compares to the present position at ~4340 m a.s.l). Similar, yet slightly less extensive advances
- were experienced by *Takdung*, *Pimu*, *Gumba*, *Dali* and *Menthosha* glaciers, which advanced ~2.41, 2.40, 2.07, 2.04 and 2.01 km, respectively (fig. 2.). Other glaciers in the *Miyar* basin probably recorded less extensive advances during this period. For example, *Tharang*, *Miyar*, *Khanjar*, and *Uldhampu* advanced by ~1.96, 1.84, 1.47and 0.69 km, respectively (fig. 2.). This magnitude, trend, and style are likely due to varying response of glaciers to local topographic factors. In terms of modern terminus elevations, *Miyar* glacier is located at the lowest elevation, terminating at ~4060 m a.s.l., followed by *Chhudong* at
- ~4343 m a.s.l. However, during the Historical Advance, the glacier with the lowest terminus elevation was *Chhudong*, at ~3931m a.s.l., followed by *Tharang* (~3939 m a.s.l.) and *Miyar* (~3973 m a.s.l.). Such varied response again suggests topographic controls on glacier advance, as the large glaciers like *Miyar* show relatively limited changes in extent when compared to medium and small glaciers.

### 4.2 Timing of human occupation at *Tharang*

The study identified and mapped three sites of abandoned settlements (ruins) in the *Tharang* end moraine complex. The sites are named as Tharang, Phundang and Patam (fig. 3.) for their locations in the basin. The Tharang channel marks a boundary between these sites. The *Tharang* ruins are found on the northern part of the *Tharang* channel where it makes a confluence with Miyar stream (fig. 3.). These ruins are found on top of the mid Holocene end moraine deposits of Tharang glacier. The 5 Phundang ruins are also located within the confines of mid Holocene moraines but on the southern edge of the confluence of the Tharang channel and Miyar stream (fig. 3.). Patam is the southern most ruins, located along the lateral moraine of mid-Holocene age (fig. 3.). Based on three well traced Kuhls (irrigation channels marked as Kuhl-I, II, III) along with terraced fields (fig. 3. and Plate 2), we infer that, at the time of occupancy, these settlements were using irrigation for their 10 agricultural activities in outwash plains. Kuhl-I, found extending up to the open yard of Tharang ruins, brought water from Darjeyang glacier and along the highest lateral moraine in the north (fig. 3. and Plate 2). The Kuhl is well marked by a twosided stone bounding and an intervening trench (Plate 2). Before reaching the settlement, the Kuhl debouched into two sequential lakes (upper and lower); formed in the complex of recessional end moraines (fig. 3. and Plate 2). When the upper lake attained its threshold, the water overflowed to the lower lake (Plate2). A distributor channel was formed from the lower lake to feed the irrigation needs of the field further down the ridges (Plate2). The two stage lakes and a separate Kuhl

- 15 lake to feed the irrigation needs of the field further down the ridges (Plate2). The two stage lakes and a separate *Kuhl* extending up to the open yard of *Tharang* ruins indicate that the *Kuhl* was used as the source of water for both domestic and agricultural needs. *Kuhl-II* is found on the right flank of the *Tharang* channel (fig. 3.). It originates from the eastern edge of *LM-I* (lateral moraine) marked on the right flank of the *Tharang* channel (fig. 3.). This *Kuhl* flows through the valley between *LM-I* and *LM-II*, feeding *Phundang* ruins and surrounding fields by one branch and *Patam* ruins by another (figure3). *Kuhl-III* is located in the centre of outwash plain, anabranches water from Tharang channel, and feeds few
- settlements along the hummocks (fig. 3.). In terms of a radiocarbon samples extracted from these ruins, *Tharang* ruins contain the highest concentration of dated

samples, UBA-30069, UBA-30074, UBA-30075, UBA-30076, UBA-30077, UBA-30078 (Table 1), with dated material comprising bone, wood, charcoal, soil hearth, Soil (cow dung) and horn, respectively. Radiocarbon samples, UBA-30073

25 and UBA-30065, were collected from *Phundang*, and comprised wood and bone (Hearth), respectively. Finally, one charcoal (hearth) sample, UBA-30064, was collected from *Patam*. The important point to note is that all of these samples were obtained from locations within and on the end moraine complex, in comparatively close proximity to the terminus of *Tharang* glacier.

The chronology based on the dates from these ruins suggests that occupation of the settlement covered a time span of cal

30 A.D ~1170 to cal AD 1730 (fig. 5.). The oldest Radiocarbon age 838±28 yr BP (UBA-30075) was obtained from the common-room hearth of the *Tharang* ruins. This indicates that this site was occupied as early as cal AD 1168 to cal AD 1224 (fig. 5.). The charcoal collected from the hearth at *Patam* ruins (UBA-30064) has returned the second oldest radiocarbon age: 654±22 yr BP, suggesting that *Patam* and *Tharang* ruins were occupied at almost the same time. A series of six dates from *Tharang* ruins range from 489 yr BP to 108 yr BP, and one sample from the common room hearth of

5

*Phundang* ruins returns an age of 101±27 yr BP. The consistent pattern of Radiocarbon ages; 838, 489, 378, 327, 212, 123, and 108 yr BP, from *Tharang* ruins, suggests that the site was continuously occupied throughout the LIA (Table 1), as the calibrated dates span from ~1170 to 1730AD (fig. 5.). These ruins were not occupied later than the late 19<sup>th</sup> century, as these moraines, and associated ruins, were marked on the GTS map of 1870 AD (fig. 4.). The dates from these three sites indicate that the oldest and longest occupied settlements, in order or antiquity are: *Tharang*, *Patam*, and *Phundang*.

Discussions

Focused and comprehensive chronological studies on Quaternary glaciation in Himalaya and Trans Himalaya (including Tibet plateau) are available now (Bookhagen and Burbank, 2006; Chen et al., 2014; Colgan et al., 2006; Lehmkuhl and

- Owen, 2005a; Mayewski et al., 2004b; Murari et al., 2014; Owen et al., 2005; Owen, 2009; Solomina et al., 2015; Tsukamoto et al., 2002; Wanner et al., 2008b). However, such focused and combined work on late Holocene glaciation in Himalaya and Tibet Oregon is missing. Recent review studies made an attempt to generate the chronological history of this area. Xu and Yi, (2014) reviewed available dates of LIA moraines in and across Tibet plateau whereas Chen et al., (2015) synthesised the proxies of moisture/precipitation in China and surrounding and Rowan, (2016) made an attempt to review the
- geochronological evidence for the LIA glacier advance in Himalaya. These studies reported asynchronous pattern of glacier attaining their maximum extent (mostly by 1300-1600 AD) in the last millennium. Whereas historical glacial advance (18<sup>th</sup> to early 19<sup>th</sup> century) was common phenomena in Tibet. But, these studies either confined to Himalaya or Tibet Plateau, mentioning that these areas are two different topographic and climatic regimes. However, LIA timing of these areas were correlated with proxies that exist far beyond these areas and have very different climatic regimes throughout quaternary
- glacial history (Gillespie and Molnar, 1995; Mayewski et al., 2004a; Solomina et al., 2015; Wanner et al., 2008b). Such restrained synthesis provides a limited understanding of the climate chronology of this region. However, understanding of climate and glacial chronology of this region (Himalaya and Tibet) as whole is of critical importance for the water needs of the most populated regions of the world.

Figure 6 depicts the combined synthesis of the chronological dates for the last 2000 years that are available for Himalaya and

- Trans Himalaya region. The figure shows two stages of glacier advance; Neoglaciation (between 300 and 900 AD) and Little Ice Age (1300-1900) in this region. Spatially, Neoglaciation was common phenomena for Eastern and Central Himalaya, along with few records for Western Himalaya and Karakoram, whereas neoglaciation was absent beyond Karakoram (Tian Shan, Qilan Shan, Hengduan Shan, Nyainqentanglha Shan, Pamir). This probably indicates that Neoglaciation in this part of the world was result of climatic mechanism (Monsoon) that dominates in Eastern Himalaya. The chronological records for
- the last millennium suggest that LIA existed here between 1300 and 1900 rather than 1300-1600 (Rowan, 2016). Spatially the duration of LIA was short (between 1300 and 1600) in Eastern Himalaya (up to Everest) whereas beyond Middle Himalaya to Tian Shan the LIA records are found prolonged between 1300 and 1900 AD (fig.6). Just opposite to neoglaciation climatic mechanism, the LIA climatic mechanism (Westerly) was active in North Western Himalaya and Trans Himalaya (fig. 5 and fig. 6). Remarkably, LIA dates are mainly from Central Himalaya (Garhwal and Everest Himalaya)

which enforce the dominance on chronology of the whole region (Rowan, 2016; Xu and Yi, 2014). This demonstrates the future scope of LIA research in other parts of Himalaya and Trans Himalaya.

The present study contributes to the broader framework for understanding climatic variability, glacier fluctuations and human interaction during Little Ice Age in Miyar Basin, and in the North West Himalaya in general. It reveals that the *Miyar* basin experienced three episodes of glacial advance (Miyar Glacial Stage, mid Holocene Advance and Historical Advance),

- basin experienced three episodes of glacial advance (Miyar Glacial Stage, mid Holocene Advance and Historical Advance), as portrayed by well-marked lateral and terminal moraines, hummocks, trimlines and an assemblage of other glacial and paraglacial landforms. However, there is no evidence to suggest LIA glacial expansion in the study area. The study documents the presence of three ruins (fig. 3.), along with unequivocal evidence of agricultural and irrigation practices (Plate 2), spanning from cal AD ~1170 to cal AD 1730 (Table 1 and fig. 5.), and clearly indicates that this area had restricted
- glacier during this period, i.e., *Tharang* glacier terminated up-valley beyond the moraine complex (Table1 and fig. 5.). On the GTS map of 1870AD (fig. 4.) these settlements are marked as ruins, suggesting that by late 19<sup>th</sup> century they were probably abandoned due to increased precipitation (snowfall) and subsequent historical advance of *Tharang* glacier. This shift in precipitation patterns likely resulted in crop failure within the basin. The evidence mentioned by Sinha et al., (2011) provide support for this explanation, as they suggested a break in Indian Summer Monsoon precipitation during AD 1400 to
- ~1700 and an increase (active) during AD ~1700 to 2007 (fig. 5.). Similar records are reported by Thompson et al., (2000)from Dasuopu Ice cores, and there is additional research which supports the glacial advance during the 17<sup>th</sup> to 18<sup>th</sup> centuries (Liang et al., 2015; Qian and Zhu, 2002; Xu and Yi, 2014; Yang et al., 2009). The occurrence of Historical Glacial advance in the 18<sup>th</sup> century is consistent with the view that plentiful precipitation is required for glacial expansion. The comparatively reduced precipitation during the course of the LIA helps to understand why glaciers in the basin remained of
- restricted extent during the period. This point is a key factor in understanding LIA climate and settlement patterns in the basin, as under modern climatic conditions, habitation and crop agriculture are confined to areas below ~3500 m a.s.l. The continuation of a settled economy, with elaborate agriculture and rich irrigation systems, for almost ~550 years, within the *Tharang* end moraine complex (at an elevation of ~3710 m a.s.l.), where agriculture is now not possible due to the short cropping season, indicates that LIA temperatures were likely higher than present (though precipitation was reduced). These
- climatic inferences suggest that the basin experienced no glacier advance during the LIA, and is an assumption supported by the region's geomorphological record. This study and comparisons with surrounding regions suggest spatial and temporal variability in terms of LIA glacial expansion (fig. 6.). Such asynchronous behaviour within the Himalayas may be a result of its topographic character and methodological practices used in dating late Holocene glacial material. The towering topography plays a dominating role in
- controlling the climate (since elevation-related precipitation and temperature gradient are notable) of the Himalaya (Bookhagen and Burbank, 2006; Chen et al., 2008; Lehmkuhl and Owen, 2005b; Owen et al., 2005; Staubwasser, 2006).The monsoon-fed regions of Eastern Himalaya and Central Himalaya did not experienced historical advance during the 1600-1900 whereas historical advance was common phenomena in Western Himalaya, Karakoram and further North West regions (fig. 6.). Similarly, LIA records are missing for Western Himalaya and Karakoram region whereas LIA was common

phenomena for the Himalayan and Trans Himalayan region (fig. 6.). For example, Gangotri glacier in Garhwal valley (Barnard et al., 2004a; Sharma and Owen, 1996), Sonapani glacier in the Upper Chandra valley (Owen et al., 1996a, 2001) and Milam glacier in Nanda Devi (Barnard et al., 2004a) recorded glacier advance during the LIA period. Whereas, glaciers in the *Miyar* basin, occupying a rain shadow of the Pir Panjal range, did not experience the advance.

- The resolution of dates irrespective of the methods is also a matter of observation and concern (Table 2). Furthermore, it is now well acknowledged that the chronology of glacier advance in the Himalaya is limited by dating techniques (Barnard et al., 2004a; Lehmkuhl et al., 1998b; Murari et al., 2014; Owen and Derbyshire, 1998; Owen et al., 1998b, 2001, 2002b; Sharma and Owen, 1996; Solomina et al., 2015). The available records are mainly from Central Himalaya (Garhwal and Everest region) and to know the whole region climatic chronology, further research is required in other regions of the
- Himalaya. Recent work based on TL (Bali et al., 2013; Blair et al., 2005; Colgan et al., 2006; Lehmkuhl and Owen, 2005; Owen et al., 2002b; Sharma and Owen, 1996; Spencer and Owen, 2004) has improved the capability of dating non-organic material in the Himalayas, but there is always scope to refine such approaches. This is illustrated in the Garhwal Himalaya, where the OSL and CRN dates of Barnard et al.(2004b) and Sharma and Owen (1996) have notably different ranges.

## 15 Conclusion

The Himalayan glacier environments are often devoid of suitable organic material for the application of <sup>14</sup>C dating. However, there are some localities where this approach can be usefully applied. The Miyar basin, a major tributary basin of the Chandrabhaga River in Lahul Himalaya, is one such location, where archaeological evidence, in the form of ruins in the end moraine complex of Tharang glacier, preserve datable organic material. Contemporary villages in this basin are

- restricted to altitudes of 2600–3500 m a.s.l, and support agriculture by diverting melt water derived from glaciers and snow through gravity-fed channels (*kuhl*). In the absence of such systems, there would be little agriculture in this "Cold Desert" region of the Himalaya. Based on the organic material obtained from three sites, namely; *Tharang, Patam* and *Phundang*, this study indicates that settlements in the Miyar basin at altitudes as high as 3700 m a.s.l., were occupied between cal AD ~1170 and cal AD 1730 (<sup>14</sup>C dates). The continued occupancy of these settlements (with elaborate agriculture and water
- harvesting structures) for almost 550 years during the  $12^{th}$  century to  $17^{th}$  century indicates comparatively mild conditions during the LIA. These sites thrived within the end moraine complex and outwash plain of Tharang glacier; just ~ 3 km from the contemporary snout. These conditions suggest that the glacier either terminated close to its current position or further upvalley during the LIA. There are no fresh moraines remains in the frontal area of this glacier to suggest that there was any noticeable "Little Ice Age" advance unlike elsewhere in the world. As a result, we suggest that, during the LIA, this region
- was warmer (and drier) than the present, allowing a settled agriculture economy to be sustained at an altitude of ~3700 m a.s.l.

#### Acknowledgement

We are thankful to the Ministry of Environment and Forest, Government of India to provide financial support for this study.

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
