# Peer review of "Glacio-archaeological evidence of warmer climate during the Little Ice Age in the Miyar basin, Lahul Himalaya, India"

_Climate of the Past, 2016_

## Referee Comment (RC1) · Anonymous Referee #1 · 9 Dec 2016

The manuscript presents a 14C-based numerical dating case study at an archaeological site located in a proglacial environment in the Miyar Basin, Lahul Himalalya, India, which is an interesting and understudied region. According to the authors their data implies that the site was inhabited and used for agriculture during the 'Little Ice Age' (LIA). Basing on the configuration of modern settlements in the valley and a review of literature on Quaternary glaciations in High Asia, conclusions regarding the paleoclimatic conditions are drawn, i.e. that the region has experienced warmer and drier climate during the LIA. Since the LIA is widely considered to be a phenomenon of world-wide impact, 'evidence' for a local or even regional climatic optimum during this period –as suggested by the title– would be of outstanding relevance for both the glaciological and

the paleoclimatic research communities. However, unfortunately the manuscript and data cannot meet such high expectations and are subject to range of substantial shortcomings as outlined below. As a consequence, I cannot recommend the manuscript for publication.

Specific comments

My criticism focuses on six points which I will explain in further detail below: (i) Lack of clear concept and study design (ii) Literature work does not fulfill basic standards (iii) Method set is too limited for the overall purpose, uncertainties are not discussed (iv) Large parts of the discussion are biased and/or speculative (v) Conclusions are not supported by data (vi) Figures are poor

In the following I will focus on the key shortcomings – not only to explain my recommendation to reject the manuscript but also to provide the authors as detailed information as possible on what I think would be necessary to improve future submissions. I did my best to present my criticism in a constructive way but am afraid much of it still won't read nicely.

(i) Lack of clear concept and study design The introduction hardly points out the relevance of the topic. Furthermore, basics of scientific study design such as hypothesis, research questions and aims do not become clear. In consequence, much of the article, particularly the discussion, left an rather incoherent impression owing to a lack of focus (cf. iv) and, ultimately, lead to conclusions which are hardly supported by the data (cf. v).

The study mixes elements of a review with elements of a case study in a way which I find rather inappropriate. The actual data should be considered in far more detail, in my opinion. Specifically, the discussion of the dating results is very limited, uncertainties are missing completely, and not much information is provided regarding spatial distributions (cf. iii). On the other hand, review elements are mostly used to fill gaps in the manuscript's dataset. Basically, there is nothing wrong with such an approach but it
certainly requires humbleness, respect for and detailed consideration of other authors' work, particularly when challenging or even neglecting a widely accepted current state of research. This could all be done in this manuscript (cf. ii).

The title is misleading since the actual dataset is neither sufficient to constrain LIA glacier extents nor related paleoclimate in my opinion (cf. iv, v).

(ii) Literature work Many statements which obviously do not represent original research results of this manuscript are lacking references. In this respect, the chapters 2 (study area) and 3.1 (mapping methods) require specific highlighting, both not citing a single reference.  Other statements which are either not the key focus of the paper (e.g. Pleistocene glaciations, P1L22ff, P7L9ff) or of limited informative value (e.g. P1L25ff) are supported by a wealth of literature. In these cases, the reader has hardly any added value and might get the impression that the citations are just to fill up the references list.

In general, I strongly suggest treating previous work with much more humbleness and respect.  Particularly in contexts regarding the LIA in High Asia the manuscript tends to reject the current state of research; However, without treating any study in detail or providing a discussion that gets close to supporting such point of view. For example at P1L29ff eleven (!)  studies, many of which presenting high quality original results, are jointly accused of falsely generalizing LIA conditions from other parts of the world over the Himalaya without going into any further detail. In my opinion, this is not justified skepticism against past studies or challenging paradigms but simply insufficient consideration of the current state of research. In this context, the treatment of a review paper by Ann Rowan (2016) needs to be highlighted: despite being up-to-date and focusing on key topics of this manuscript, the article gets only very limited attention and, whenever it is mentioned, there is always a negative annotation without explaining why. In my opinion, good scientific practice requires to work in detail with such literature, exactly covering the manuscript's topic and wider study region. Rejecting relevant studies certainly requires discussing them in detail and trying to falsify them basing on actual

data – one by one.

Also, much key literature is missing, e.g. regarding dendrochronological studies of LIA glaciations and Holocence climate, the heterogeneity in climatic forcing of glacier dynamics over High Asia, existing morphostratigraphies, etc. (cf. literature recommendations at the end of this text).

(iii) Method set and uncertainties CRN and OSL dating have made great progress during the last years, CRN is certainly the most promising technique for LIA moraines without tree stands today. The literature used to support the weak and vague argument that these methods have "limitations" (P2L15ff) seem either outdated or inappropriate for the context of LIA glaciers.

By contrast, a critical assessment of the 14C method or discussion of uncertainties is lacking. The plateau in the 14C calibration curve, one of the main reasons why 14C is used so scarcely in LIA studies because it renders calibration almost impossible, is not even mentioned.

Dendro methods are key for both paleoclimate and glacier characterization in the LIA context. These are also not even mentioned in the manuscript.

The assumption that the area was used for agriculture should be supported by sedimentological and pedological analyses.

(iv) Lack of focus as well as biased and/or speculative discussion Even though the manuscript is quite short, much text is used for off-topic statements. A prominent example is the first paragraph of the results (P4L27-P5L2) which basically presents no results at all. Instead, again the LGM configuration is highlighted (without citing adequate literature or indicating how this is relevant for the study at hand). The next paragraph (P5L3-P5L12) presents "middle Holocene" moraines without providing any dating evidence (and so on).

The data actually presented is restricted to 14C dating results which are accompanied

by some archaeological interpretations of landforms and landform mapping. Paleoclimatic datasets from the study area are not part of this paper.

The discussion is extremely speculative in my opinion, in some parts completely losing the connection to the data and/or the literature which is actually available in the study region, at times even to the topic of the paper.

The author's conception of the term "Little Ice Age" is not congruent with widely accepted definitions from the literature. Despite the typical fuzziness regarding the determination of starting and end points of certain periods (mostly originating from different events and/or dating methods used for definition) I would say that the LIA is generally used for a climatic pessimum lasting from ∼ 1300 until ∼1900 CE. Glacier advances in the 19th century glacier advances would thus be considered (late) LIA by most authors. The term 'historical glacier advance' which the authors use instead is typically used when referring to glacier fluctuations for which actual observational data exists. Despite different local dynamics and timings, the LIA cooling signal is regarded to be world-wide owing to the global character of predominant forcing, i.e. solar and volcanic. As such, supporting the hypothesis of a warming during LIA must be substantiated by outstandingly solid data.

On the other hand, the 'Medieval Warm Period' (MWP, also referred to as 'Medieval Climate Anomaly') lasted ∼1000-1300 CE. The oldest sample (UBA-30075) is thus definitely MWP/pre-LIA, the second oldest (UBA-30064) may arguably be considered MWP as well but that will depend on actual data on late Holocene climate in the study area.

Figure 2 shows a clear LIA moraine in my opinion (lobate structure just left below the center of the Google Earth image) – with both tributaries of Tharang Glacier united as indicated in the historical map. The historical map, on the other hand, has a much to coarse resolution to use it as a basis to constrain the actual terminal position. CRN dating of this moraine will probably be the only way to assess the timing.

(v) Conclusions are not supported by data I certainly agree that the mapped structures and archaeological findings indicate certain farming usage. However, to me much evidence (horns, cow dung) seems to point toward pastures rather than agriculture. Pasturing has yet much less climatic implications since pasturing has been the predominant type of land usage in High Asia for thousands of years despite climate fluctuations. In order to support the argument that actual agricultural activities were conducted at the sites more evidence needs to be presented, e.g. soil profiles with substantial portions of humic matter, drill furrows, pollen records.

(vi) Figures Complex, confusing and cluttered figures that are neither self explaining nor being explained in figure captions. Linkages between different figures (e.g. marking locations of photos and map in overview) as well as figures and text are weak or do not exist.

Literature suggestions Bräuning, A., 2006. Tree-ring evidence of Little Ice Age glacier advances in southern Tibet. The Holocene 16, 369–380. doi:10.1191/0959683606hl922rp

Grießinger, J., Bräuning, A., Helle, G., Thomas, A., Schleser, G., 2011. Late Holocene Asian summer monsoon variability reflected by $\delta$18O in tree-rings from Tibetan junipers. Geophysical Research Letters 38, 1–5. doi:10.1029/2010GL045988

Kotlia, B.S., Ahmad, S.M., Zhao, J.-X., Raza, W., Collerson, K.D., Joshi, L.M., Sanwal, J., 2012. Climatic fluctuations during the LIA and post-LIA in the Kumaun Lesser Himalaya, India: Evidence from a 400 ǎy old stalagmite record. Quaternary International, Late Quaternary morphodynamics in East Asia 263, 129–138. doi:10.1016/j.quaint.2012.01.025

Krusic, P.J., Cook, E.R., Dukpa, D., Putnam, A.E., Rupper, S., Schaefer, J., 2015. Six hundred thirty-eight years of summer temperature variability over the Bhutanese Himalaya. Geophys. Res. Lett. 2015GL063566. doi:10.1002/2015GL063566

Liang, F., Brook, G.A., Kotlia, B.S., Railsback, L.B., Hardt, B., Cheng, H., Edwards, R.L., Kandasamy, S., 2015. Panigarh cave stalagmite evidence of climate change in the Indian Central Himalaya since AD 1256: Monsoon breaks and winter southern jet depressions. Quaternary Science Reviews 124, 145–161. doi:10.1016/j.quascirev.2015.07.017

Schaefer, J.M., Denton, G.H., Kaplan, M., Putnam, A., Finkel, R.C., Barrell, D.J.A., Andersen, B.G., Schwartz, R., Mackintosh, A., Chinn, T., Schlüchter, C., 2009. High-Frequency Holocene Glacier Fluctuations in New Zealand Differ from the Northern Signature. Science 324, 622–625. doi:10.1126/science.1169312

Schimmelpfennig, I., Schaefer, J.M., Akçar, N., Koffman, T., Ivy-Ochs, S., Schwartz, R., Finkel, R.C., Zimmerman, S., Schlüchter, C., 2014. A chronology of Holocene and Little Ice Age glacier culminations of the Steingletscher, Central Alps, Switzerland, based on high-sensitivity beryllium-10 moraine dating. Earth and Planetary Science Letters 393, 220–230. doi:10.1016/j.epsl.2014.02.046

Yadav, R.R., Braeuning, A., Singh, J., 2011. Tree ring inferred summer temperature variations over the last millennium in western Himalaya, India. Clim Dyn 36, 1545–1554. doi:10.1007/s00382-009-0719-0
* * *

---

## Short Comment (SC1) · 1 Mar 2017

We thank the reviewer for the comments that helped us to improve the manuscript. We have incorporated the comments wherever found applicable. We have particularly refined the introduction, discussion with more research papers and provided more detailed analysis. We reply to the comments as the following. Answer should be read as mentioned in the modified manuscript.

1. **Anonymous Referee #1**

*1.1 The introduction hardly points out the relevance of the topic. Furthermore, basics of scientific study design such as hypothesis, research questions and aims do not become clear.*

*1.2 The study mixes elements of a review with elements of a case study in a way which I find rather inappropriate.*

*1.3 The title is misleading since the actual dataset is neither sufficient to constrain LIA glacier extents nor related paleoclimate in my opinion.*

**Answer**

1.1 The Introduction has been modified to make it clearer to understand the relevance of the topic, hypothesis, research questions and aims.

Based on the quaternary and Holocene asynchronous pattern of Himalayan glacier, the study investigates on the hypothesis that Himalayan glaciers are asynchronous to Northern Latitude glaciers (European glaciers). The basic research question which is addressed is that when the period of rapid climate change (LGM and Holocene) did not experience synchronicity between Himalayan and northen latitude glaciers, how the period of **none-rapid climate change (LIA)** can have similarity in glacier response.  On the part of the relevance and aims of the study; it is clearly mentioned in the introduction that uncertainties persist regarding the timing extent and causes of LIA at regional and global scales. As the available palaeo-climatic proxies for LIA in Indian subcontinent suggest weaker (Thompson et al., 2000; Gupta et al., 2003; Sinha et al., 2011; Yadav, 2011) as well as stronger monsoon (Liang et al., 2015) during the LIA. Therefore, it provides opportunity to examine the discrepancies in the palaeo-climatic proxies across the Himalaya. The study, evaluates the glacio–archaeological feature in the Miyar Basin, Lahul Himalaya, using landform evidence and radiocarbon dating techniques.

1.2 The discussion has been modified and now the review studies and cases studies are separately dealt in different paragraphs and the synthesis is made in one to one cases with proper citations. As

"A focused and combined work on late Holocene glaciation in Himalaya and Tibet orogen is rather difficult to find. However, recent review studies have attempted to generate the chronological history of this region. Xu and Yi, (2014) reviewed available dates of LIA moraines in and across Tibet plateau whereas Chen et al., (2015) synthesised the proxies of moisture/precipitation in China and surrounding regions. Rowan (2016) has provided a review of the geo-chronological evidence for the LIA glacial advances in the Himalaya. Based on the review works of  Xu and Yi (2014) and Rowan (2016), Figure 6 presents a combined synthesis of dated chronologies for the last 2000 years available for the Himalaya and Trans Himalayas. It shows two stages of glacier advance i.e. Neoglaciation (between 300 and 900 AD) and Little Ice Age (1300-1900) in this region. Neoglaciation was common

phenomena for Eastern and Central Himalaya, along with few records for the Western Himalaya and Karakoram, whereas it was absent beyond Karakoram (Tian Shan, Qilan Shan, Hengduan Shan, Nyainqentanglha Shan, Pamir). Such spatial pattern indicates that Neoglaciation was result of different climatic mechanisms that dominate in the Western and Eastern Himalaya. The chronological records for the last millennium suggest that peak of LIA existed between 1300 and 1900 rather than 1300-1600 (Rowan, 2016) in the region. However, duration of LIA was shorter (between 1300 and 1600) in the Eastern Himalaya (up to Everest) than the Middle and Western Himalaya and beyond (1300 and 1900 AD) (fig.6). Contrary to extended Neoglaciation fluctuations in Eastern and Central Himalaya, the LIA was relatively more active in the North Western and Trans Himalaya (fig. 5 and fig. 6). Noticeably, the frequency of obtained LIA dates is mainly from Trans Himalaya (beyond Pamir) and Garhwal and Everest Himalaya which enforce the dominance on chronology for the entire region (fig.6).

Available case studies have described the LIA glacial advance in the parts of the Himalaya (Owen et al., 1996a, 1996b; Sharma and Owen, 1996; Derbishyre and Owen, 1997; Lehmkuhl et al., 1998; Owen et al., 1998b, 2000; Taylor and Mitchell, 2000; Owen et al., 2001, 2002a, 2002b; Spencer and Owen, 2004; Owen et al., 2005; Hedrick et al., 2011; Murari et al., 2014; Solomina et al., 2015). However, these studies vary in terms of timing and extent of glacial advance, and most are based on relative rather than absolute chronologies (Mayewski and Jeschke, 1979; Owen et al., 1996a, 1996b; Sharma and Owen, 1996; Derbishyre and Owen, 1997; Lehmkuhl et al., 1998; Owen et al., 1998b, 2000; Taylor and Mitchell, 2000; Mehta et al., 2012).Wherever the numerical dating has been applied, there are only a limited number of dates ($\leq 2$) (Iwata, 1976; Richards et al., 2000; Owen et al., 2001). In areas where more dates have been produced such as Khumbu (Bendict, 1976; Iwata, 1976; Fushimi, 1978; Muller, 1980; Rothlisberger and Geyh, 1986; Richards et al., 2000), Garhwal (Barnard et al., 2004b; Murari et al., 2014), Milam (Barnard et al., 2004a), Gonga Shan (Owen et al., 2005), most are based on thermo luminescence (TL) and optically-stimulated luminescence (OSL) dating techniques. Though these methods are widely applied globally, they have associated limitations in high energy Himalayan environment in determining ages of such events within millennial scale (Jensen et al., 2000; Wallinga, 2002; Spencer and Owen, 2004; Blair et al., 2005; Bailiff et al., 2014). However, we have used the exposure and luminescence dating techniques where confidently sampled in the same environmental settings and geographical conditions ( Deswal et al., 2012)."

1.3 We differ from the reviewer's comment as the present study produce 9 radiocarbon dates based on glacio-archaeological remains along with extensive landform mapping, along with field photographs. The number of produced dates in this study is one of the highest among the published case studies regarding the LIA in the Himalayas.

However, we partially modify the manuscript title as "**Glacio-archaeological evidence of climate during the Little Ice Age in the Miyar basin, Lahul Himalaya, India**".

**2. Anonymous Referee #1**
*2.1 Literature work Many statements which obviously do not represent original research results of this manuscript are lacking references. In this respect, the chapters 2 (study*

*area) and 3.1 (mapping methods) require specific highlighting, both not citing a single reference.*

*2.2 Other statements which are either not the key focus of the paper (e.g. Pleistocene glaciations, P1L22ff, P7L9ff) or of limited informative value (e.g. P1L25ff) are supported by a wealth of literature. In these cases, the reader has hardly any added value and might get the impression that the citations are just to fill up the references list…. also, much key literature is missing, e.g. regarding dendrochronological studies of LIA glaciations and Holocene climate, the heterogeneity in climatic forcing of glacier dynamics over High Asia, existing morphostratigraphies, etc. (cf. literature recommendations at the end of this text).*

**Answer**

2.1 We have incorporated your comments regarding the citation for study area and mapping.

2.2 for the comments we justify that this was incorporated in the study as the formulation of hypothesis of the study is based on the behaviour of Himalayan glacier during the quaternary and Holocene, therefore citation of such behaviour is must including LGM and Holocene.

In the modified manuscript, we have included more paleoclimatic proxies including dendrochronology, speleothem etc.

**3. Anonymous Referee #1**

*3.1 Method set and uncertainties CRN and OSL dating have made great progress during the last years, CRN is certainly the most promising technique for LIA moraines without tree stands today. The literature used to support the weak and vague argument that these methods have "limitations" (P2L15ff) seem either outdated or inappropriate for the context of LIA glaciers.*

*3.2 By contrast, a critical assessment of the 14C method or discussion of uncertainties is lacking.*

3.3 Dendro methods are key for both paleoclimate and glacier characterization in the LIA context. These are also not even mentioned in the manuscript.

3.4 The assumption that the area was used for agriculture should be supported by sedimentological and pedological analyses.

**Answer**

**3.1** The discussion paragraph (page8; line 6-17) deals with this answer. The LIA "studies vary in terms of timing and extent of glacial advance, and most are based on relative rather than absolute chronologies (Mayewski and Jeschke, 1979; Owen et al., 1996a, 1996b; Sharma and Owen, 1996; Derbishyre and Owen, 1997; Lehmkuhl et al., 1998; Owen et al., 1998b, 2000; Taylor and Mitchell, 2000; Mehta et al., 2012).Wherever the numerical dating has been applied, there are only a limited number of dates ($\leq 2$) (Iwata, 1976; Richards et al., 2000; Owen et al., 2001). In areas where more dates have been produced such as Khumbu (Bendict, 1976; Iwata, 1976; Fushimi, 1978;

Muller, 1980; Rothlisberger and Geyh, 1986; Richards et al., 2000), Garhwal (Barnard et al., 2004b; Murari et al., 2014), Milam (Barnard et al., 2004a), Gonga Shan (Owen et al., 2005), most are based on thermo luminescence (TL) and optically-stimulated luminescence (OSL) dating techniques. **Though these methods are widely applied globally, they have associated limitations in high energy Himalayan environment in determining ages of such events within millennial scale (Jensen et al., 2000; Wallinga, 2002; Spencer and Owen, 2004; Blair et al., 2005; Bailiff et al., 2014). However, we have used the exposure and luminescence dating techniques where confidently sampled in the same environmental settings and geographical conditions ( Deswal et al., 2012)."**

*3.2* The answer is dealt in **3**. **Methodology** heading **3.2.2 Laboratory Work** (page4; lines22-24): "The obtained radiocarbon dates were calibrated in the CALIB RADIOCARBON CALIBRATION PROGRAM 1986-2016 using IntCal13 calibration curves (Reimer, 2013). The uncertainties for the calibrated ages are given up to 1σ (Table1)." Further to this the dates had been discussed in detail in **Results; 4.2 Timing of human occupation at** *Tharang (pages 6-7, lines 27-34, 1-7 respectively):*
"The chronology of these ruins was established based on the 9 dates extracted from these ruins; comprising bone, wood, charcoal, soil (hearth), soil (cow dung), soil (toilet) and horn (Table1). *Tharang* ruins contain the highest concentration of dates (7 samples), UBA-30069, UBA-30072, UBA-30074, UBA-30075, UBA-30076, UBA-30077, UBA-30078 whereas, UBA-30064 and UBA-30065 were collected from *Patam* and *Phundang ruins* respectively (Table 1). All these samples were obtained from sites within and on the end moraine complex, in comparatively close proximity to the terminus of *Tharang* glacier (4.1 Km). The seven dates of *Tharang* ruins are 838 ±28, 489 ±22, 378 ±27, 327 ±21, 212 ±34, 123 ±22, and 108 ±32 yr BP (Table1). The consistent pattern of Radiocarbon ages; ranging from 838 yr BP to 108 yr BP, from *Tharang* ruins, suggests that the site was a continuous living village between cal AD 1168 and cal AD 1693. As the oldest Radiocarbon age (UBA-30075) 838 ±28 yr BP, at 68% confidence level (1σ uncertainty) has an average age of 1196 AD, spanned from cal AD 1168 to cal AD 1224. Whereas the latest age (UBA-30072); 108 ±32 yr BP, at 68% confidence level (1σ uncertainty) has an average age of 1710 AD, spanned from 1693-1727 AD. Moreover, there are five dates in between; and at 68% confidence level, the average of these five dates are 1430, 1476, 1522, 1664, and 1696 (Table1). Besides, the dates of Tharang, *Patam* ruins received the second oldest radiocarbon age: 654 ±22 yr BP (UBA-30064), with an average age of 1297 yr BP, spanned from cal AD 1289 to cal AD 1305 whereas, *Phundang* ruins received most recent date (UBA-30065) 101 ±27. The date spanned from cal AD 1695 to cal AD 1726, with an average age of 1711".

3.3 Dendrochronology for the rain shadow zone of Pir Panjal (Lahul Himalaya) is merely available, however the dendrochronology south of of Pir Panjal has been incorporated in the discussion but represents to a very different climatic system (Monsoon Active zone).

3.4 The answer is dealt in the modified manuscript under Results heading **Timing of human occupation at *Tharang*** *(page 6 lines 5-12).*

**4. Anonymous Referee #1**

*4.1 Lack of focus as well as biased and/or speculative discussion…. the LGM configuration is highlighted (without citing adequate literature... The next paragraph (P5L3-P5L12) presents "middle Holocene" moraines without providing any dating evidence.*

*4.2 The author's conception of the term "Little Ice Age" is not congruent with widely accepted definitions from the literature….. The author's conception of the term "Little Ice Age" is not congruent with widely accepted definitions from the literature. Despite the typical fuzziness regarding the determination of starting and end points of certain periods (mostly originating from different events and/or dating methods used for definition) I would say that the LIA is generally used for a climatic pessimum lasting from ~ 1300 until ~1900 CE. Glacier advances in the 19th century glacier advances would thus be considered (late) LIA by most authors. The term 'historical glacier advance' which the authors use instead is typically used when referring to glacier fluctuations for which actual observational data exists…*

*4.3 On the other hand, the 'Medieval Warm Period' (MWP, also referred to as 'Medieval Climate Anomaly') lasted ~1000-1300 CE. The oldest sample (UBA-30075) is thus definitely MWP/pre-LIA, the second oldest (UBA-30064) may arguably be considered MWP as well but that will depend on actual data on late Holocene climate in the study area.*

*4.4 Figure 2 shows a clear LIA moraine in my opinion (lobate structure just left below the center of the Google Earth image) – with both tributaries of Tharang Glacier united as indicated in the historical map.*

**Answer**

**4.1** The modified results under the heading **Glacier Advances and Landforms (page 4-6) refers the relevant literature for LGM and the dates of the Holocene advance (**dated to ~10 ±1.0 ka to 8 ±1.0 ka OSL), terminal moraine of Tharang has yielded 6.6 ±1.0 ka (Deswal, 2012) is well cited.

As far as the speculative discussion comment is concern, we differ from the comment; however more literature has been cited for the argument in the discussion; including dendrochronological, speleothem, and ocean oxygen isotope records which are available across the Indian subcontinent.

**4.2** We never mentioned that LIA is not considered between 1300-1900 CE. However, there is spatial variability regarding extent and timing across the Himalaya (fig. 6). Moreover, the connotation of the Historical Advance is made on the basis of the available GTS map (1870s fig. 4) and Harcourt's map (1871) of the study area well mentioned in the results and discussion.

**4.3** we agree with the comment and we have specifically mentioned in the results that these two dates belong to MWP.

**4.4** The figure2 moraine are marked on the historical map (1874) are not of LIA advance rather belong to Historical Advance only (after 1730) as three ruins settlement along with well-developed irrigation system survived between 1168-1730 (table1) in the end moraine complex of the same glacier.

**5. Anonymous Referee #1**

*Conclusions are not supported by data I certainly agree that the mapped structures and archaeological findings indicate certain farming usage. However, to me much evidence (horns, cow dung) seems to point toward pastures rather than agriculture.*

**Answer**

The modified conclusion on page 10 concludes on the basis of the 9 radiocarbon dates and landform interpretation. As far as the pasture activities is concern the explanation is made in the **Results** under the heading **4.2 Timing of human occupation at *Tharang* (page 6; lines 11-12).**

**6. Anonymous Referee #1**

*Figures Complex, confusing and cluttered figures that are neither self-explaining nor being explained in figure captions. Linkages between different figures (e.g. marking locations of photos and map in overview) as well as figures and text are weak or do not exist.*

**Answer**

We take this comment positively for the betterment of the manuscript and have updated some figures which were necessary.

---

## Referee Comment (RC2) · Anonymous Referee #2 · 6 Mar 2017

The main goal of this ms is to present detailed mapping of ruins within a moraine complex high in the Miyar basin in the Indian Himalaya, and to show that new 14C dates from the ruins indicate continuous human occupation from 1170-1730 AD. As the moraine complex is 200 m above the highest cultivation at present, the authors conclude that their data demonstrate that the LIA there was warmer than present. This would be an unusual conclusion.

The research team has clearly done detailed mapping within the moraine complex and documentation of the ruins, the crop fields and the irrigation techniques used by the former inhabitants. This looks very solid. However, I have some reservations on the format of the ms.

[Figure]

I have several questions and some suggestions on how to improve the ms.

Continuous line numbers for the ms would be very helpful, rather than starting over every page

Abstract

Avoid passive text "is described" and superlatives "Impressive"

Organization

The ms has elements of a major review of Holocene glaciation in the Himalaya, and also elements of a very detailed local study which is, I think, the major contribution. I suggest the authors remain focused on their own study and limit overview text to their own region. An overview of what is known of the LIA in general for the Himalaya would be fine. I have suggested below an alternative organizational structure.

Geochronology

The main focus of my concern is the dating, which is the key element of their contribution. Given the difficulty of locating appropriate organic material for dating, it is good to see some new ages. However, the authors' treatment of the dates needs to be expanded. Table 1 needs to include the Conventional 14C ages and the Calibrated Ages. They have not correctly calibrated their dates. It is simply not possible to have the very small 1 sigma uncertainties for ages less than 400 14C years that are listed in Table 1; this is very clear from their own Figure 5, where they appear to have included the results of the calibration, showing vastly greater uncertainties for the younger samples.

I suggest the authors provide details of the collection for each of the radiocarbon samples. This is not given in the text, and also how they interpret them. Was the laboratory able to extract collagen from the bones? Are the horns and bones collected on the surface or in stratigraphic section? If from the surface do they represent casual animal presence?

The large discrepancy between the hearth charcoal dates and the animal dates is thought provoking. Where did the inhabitants obtain wood for fires? Is it "fossil" wood. . .that is wood that is long dead and so has an "old" 14C age? Are the charcoal dates on a single discrete piece of charcoal or on many pieces?

How do they interpret their "soil" dates? I assume these are "bulk" dates as opposed to dating a discrete item. Soils are always time integrators.

What is the context of the dated wood? Is it part of the structures of the ruins, or from or just lying on the surface? Why is it there? Was it to be used for firewood?

Very little information is provided on these important aspects, which the reader requires as the primary conclusions are all based on the 14C dates.

Line 26-28 p. 6 " The important point to note is that all of these samples were obtained from locations within and on the end moraine complex, in comparatively close proximity to the terminus of Tharang glacier." From their Fig. 2 it seems to me that the end moraine complex containing the ruins is far from the terminus of the glacier, in fact from Fig. 2 it is 5 km and around the corner (although in the conclusions they say it is only 3 km to the snout; which is it?)

Lines 4,5 p.7 "The dates from these three sites indicate that the oldest and longest occupied settlements, in order or antiquity are: Tharang, Patam, and Phundang. " is certainly not justified from the data. There is only one date listed from both Patam and Phundang, although the text on p. 6 says two of the dates are from Phundang. It is not possible to draw such a conclusion.

I suggest giving all 14C ages (Conv and Cal) in years AD; it will be much easier to follow the text that way,

Discussion

This lengthy section includes a long discussion that while interesting seems more of a review than of relevance to their primary dataset and the focus of this paper.

A few key points:

Neoglaciation has a formal definition and it includes the LIA. The authors must change their terminology. The advance called the "Historical Advance" within the interval the authors state is the LIA, but is nevertheless separated from the LIA in their tables. This makes no sense to me. They discuss periods of both greater and lesser precipitation during the LIA on p.8 yet say the LIA had little glacier response because there was less precipitation even though there was apparently greater precipitation from 1700-1900 AD, clearly during what is still the LIA. I think the authors must consider the "Historical Advance" to be a LIA phenomenon.

The authors argue that the moraine in which the ruins are situated in from middle Holocene. This might be true, but is there any evidence for a 6 ka age? How do they know it is not a recessional moraine from regional deglaciation at the end of the last ice age? The ruins provide only a minimum age. Their statement (p.8) " The study . . ... clearly indicates that this area had restricted glacier during this period [occupation], i.e., Tharang glacier terminated up-valley beyond the moraine complex" seems oddly out of place as they had already argued it was mid Holocene in age and had not cited anyone else arguing it was LIA in age. They go on to argue that an increase in precipitation in the late 19th Century caused crop failure (p.8), which is also odd, since the region is moisture limited for crops. It would have to be either less precipitation or colder summers to cause crop failure.

Conclusions

This should be a summary of their key findings (the dates and what they mean) and the climate interpretations. Their statement " the glacier either terminated close to its current position or further up- valley during the LIA" seems not to be supported by their Fig 2 that indicates an "Historical (which is LIA) Advance", but that figure is very difficult to understand. Finally, the authors imply in the conclusions that the dominant control on the mass balance of Tharang Glacier is summer temperature, when in this region

precipitation has to be a major contributor.

Figures: Captions need far more explanation.

Fig. 1 Not possible to see where the field area is or what the small inset figures mean

Fig. 2 requires a lot of guess work; not possible to fully understand; two advances mentioned but not clearly labeled which is which

Fig. 3 OK

Fig. 4 Not particularly helpful

Fig. 5 OK (NH Temp Anomaly not cited. The sites of these records need to be located somehow. The Dasuopu record seems most relevant and suggests precip control on Tharang glacier?

Fig. 6 should not be included. Save this for a review paper. It's not possible to interpret it in its current format

Plates are not helpful. I was hoping to get a sense of the 14C samples but you can't tell from the plates

Recommendation

The team has clearly done a lot of work in the Tharang Glacier region. However, in its current form, the ms is not appropriate for Climate of the Past. All the conclusions are derived from the 14C dates, but the calibrations as given in Table 1 are incorrect, which causes incorrect conclusions. A revision could be acceptable if the authors can provide more secure detail on the stratigraphic settings for each of their dates, correctly calibrate them (especially the true uncertainties) and consider the balance between precipitation and temperature on the glacier mass balance. They need to accept that any advance before 1900 AD and after 1250 AD should probably be considered a LIA advance. They should present evidence for the age of the moraine on which the ruins are located...are they sure it is a mid Holocene advance or could it be an LGM

recessional moraine. Not too important for their main story, but if there is evidence that it is 6 ka, this would be of interest to the readership.

Possible alternative Outline for a revised paper

Abstract

Introduction

Study area location and climate

Methods

Field Mapping

Ruins

Glacial Features

Radiocarbon Dating

Results

Holocene glaciation of the Miyar Basin

Radiocarbon Dates

Details of each collection

Interpretation of each date

Interpretation

Human occupation of the moraines: This should be a discussion on how robustly can they conclude continuous occupation of the ruins by humans. What is age of this moraine?

Climate control on late Holocene glaciation: Here we need a discussion of the balance between precipitation and melt to control glacier mass balance. Is the lack of a LIA

advance due to aridity and NOT warmer temperatures? What do we actually know about Tharang Glacier mass balance? Is it likely more sensitive to precipitation or temperature?

Discussion

I think focus on LIA evidence from the nearest regions

Conclusions